# Dual Mode Gait Sonification for Rehabilitation After Unilateral Hip Arthroplasty

**DOI:** 10.3390/brainsci9030066

**Published:** 2019-03-19

**Authors:** Julia Reh, Tong-Hun Hwang, Gerd Schmitz, Alfred O. Effenberg

**Affiliations:** Institute of Sports Science, Leibniz University Hannover, Am Moritzwinkel 6, 30167 Hannover, Germany; tonghun.hwang@sportwiss.uni-hannover.de (T.-H.H.); gerd.schmitz@sportwiss.uni-hannover.de (G.S.); effenberg@sportwiss.uni-hannover.de (A.O.E.)

**Keywords:** gait sonification, acoustic feedback, hip arthroplasty, real-time sonification, acoustic model, rehabilitation, motor relearning

## Abstract

The pattern of gait after hip arthroplasty strongly affects regeneration and quality of life. Acoustic feedback could be a supportive method for patients to improve their walking ability and to regain a symmetric and steady gait. In this study, a new gait sonification method with two different modes—real-time feedback (RTF) and instructive model sequences (IMS)—is presented. The impact of the method on gait symmetry and steadiness of 20 hip arthroplasty patients was investigated. Patients were either assigned to a sonification group (SG) (*n* = 10) or a control group (CG) (*n* = 10). All of them performed 10 gait training sessions (TS) lasting 20 min, in which kinematic data were measured using an inertial sensor system. Results demonstrate converging step lengths of the affected and unaffected leg over time in SG compared with a nearly parallel development of both legs in CG. Within the SG, a higher variability of stride length and stride time was found during the RTF training mode in comparison to the IMS mode. Therefore, the presented dual mode method provides the potential to support gait rehabilitation as well as home-based gait training of orthopedic patients with various restrictions.

## 1. Introduction

Sensory feedback is of fundamental importance for motor learning and re-learning in rehabilitation [1]. An expansion of perception beyond habitual sensorimotor feedback (i.e., augmented feedback) has the potential to improve and accelerate the rehabilitation process following various diseases of the locomotor system [2,3,4,5]. Different sensory feedback systems have been developed for such purposes and have already been used in sports and rehabilitation [6,7,8,9]. Usually, visual feedback is preferred as it addresses our predominant sense and might be easier to follow, as visually perceived motion in space is unambiguously assigned and represented in the human brain [10,11,12]. Vision, however, is often involved in environmental perception, which limits the potential applications of visual feedback systems for free locomotion training [13,14,15,16]. In this respect, auditory feedback can be considered as an alternative, particularly related to cyclic movements, as the human auditory system is very sensitive in perceiving meter, rhythms, and time-dependent variations [17,18,19]. For this reason, using sound to impact movements or setting motion to music might be beneficial, specifically for cyclic movements such as walking, stair climbing, running, cycling, or swimming. There are already several studies investigating the impact of sound, rhythm, and sonification on movement types like this [20,21,22,23,24].

Using sonification to improve or optimize movements can be implemented in different ways. While some studies used real-time feedback, others have shown effects of acoustic error feedback or cueing movements [25,26,27,28,29,30]. A distinction can be made between these three approaches (real-time feedback, acoustic error feedback, and instructing or cueing movements). Real-time feedback reflects a movement acoustically. Kinetic or kinematic data are measured and mapped to a specific sound by a function. Thus, a movement causes an immediate change or onset of the related sound, which therefore is directly influenced and created by the user [31,32,33]. Compared with this, acoustic error feedback defines a certain measuring range of a kinematic or dynamic parameter (e.g., range of motion of the elbow or knee joint, weight loading of the feet) and compares measured motion data to these (also labeled as ‘bandwidth-feedback’) [25,26]. A sound signal is only played, if a given threshold is crossed. Additionally, instructing or cueing movements means that a specific predefined sound, like the ticking of a metronome, sets a temporal structure. Consequently, movements can be aligned to the sound, but the sound cannot be influenced by the user [34,35,36].

As gait is a fundamental human locomotion, it has already been subject in a wide range of studies on various types of acoustic feedback [37,38,39]. Heterogeneous study designs and various parameters were considered in these studies, as hypotheses and methods were clearly related to the limitations or diseases of the particular patients. For instance, in a pilot study Rodger et al. [40] examined the impact of two different sound approaches on gait disturbances of ten Parkinson’s disease patients. They found decreased step length variability for step cueing (first approach) and real-time feedback (second approach) compared to a no sound condition in this patient population. Additionally, Schauer and Mauritz [24] investigated musical motor feedback for stroke patients, which means that a song with an adjustable speed provided the gait rhythm. The results showed increased stride length and gait velocity as well as decreased gait symmetry in the test group. There are further pilot and feasibility studies examining various methods of acoustic feedback for gait [41,42,43], but as far as we know, only few studies on patients with orthopedic restrictions of the musculoskeletal system have been conducted [44]. Several studies examined the impact of auditory feedback on healthy persons’ gait pattern [28,39,41,45,46]. Depending on the acoustic feedback method, varying effects have been reported: Real-time sonification resulted in decreased gait speed [39,41] and decreased cadence [41], auditory cues, which were adjusted to the symmetry of the participants’ movement pattern, affected gait steadiness [45], while rhythmic music cues led to increased gait speed, stride length, and cadence [28]. However, up to now there has been no evidence as to whether auditory cues or real-time sonification can improve gait in terms of symmetry and steadiness in healthy older persons and orthopedic patients.

Patients suffering from hip or knee arthrosis frequently develop a malfunctioning gait pattern, which is often progressing over time. Patients following unilateral total hip arthroplasty (THA) commonly show asymmetric step length [47,48] causing additional long-term impairments of the musculoskeletal system. Therefore, an improved step length symmetry is an important factor in gait rehabilitation and should be considered in gait therapy. However, it is a great challenge for patients to relearn a symmetric and steady gait pattern. Usually gait rehabilitation after hip or knee arthroplasty is associated with a large effort of time and personnel [49,50]. In addition, it must be considered that prevalence of arthrosis increases with age [51,52] and thus often affects elderly patients who suffer from several co-morbidities such as cognitive impairments. In this regard, a gait rehabilitation system that does not require high attentional cost might be a powerful add-on to classical treatments.

For these reasons, we developed a new acoustic feedback approach, which is based on a combination of kinematic real-time feedback (RTF) and instructive model sequences (IMS) [53]. A consistent sound in accordance with the human gait pattern was developed and applied, based on kinematic data recorded by a portable inertial sensor system. RTF is based on selected kinematic parameters (ground contact of the feet and angular velocity of the knee joint), which are clearly mapped to a sound. This means that each ground contact and each knee extension of the patient triggers the onset, frequency, and amplitude of a defined sound with low latency (sonification). On the other hand, IMS present the same sound as used for RTF, but in a predefined manner. Consequently, IMS display acoustic information at a fixed tempo, which is comparable to cueing movements. We intended to create a close motion-sound and sound-motion linkage in terms of an establishment of co-activation patterns between the auditory and motor networks responsible for audition and motor execution [54]. To achieve a high efficiency of the method, RTF alternating with IMS was presented, enabling the calibration of the feedback to a symmetric model [55]. Furthermore, the combination of RTF and IMS complies to the theory of modularity in multisensory integration as proposed by Tagliabue and McIntyre (2014) [56]. The theory predicts optimized multisensory integration, if the movement goal is instructed in the same modality as the own movement is perceived in. Moreover, it can be assumed that orthopedic patients in general do not show any neurological limitations, in particular regarding motor planning. Therefore, this patient group may demand for especially close links between sound and motion. However, to the authors knowledge, acoustic feedback for rehabilitation in orthopedic patients has not yet been investigated. This is why the current study provides first insights into a new application.

The developed method was applied in a two-week intervention study on patients with unilateral THA in order to examine the following two hypotheses:

First, (H 1) the dual mode acoustic feedback method improves the patients’ gait symmetry over two weeks compared to a control group without acoustic feedback. Second, (H 2) the effects on the gait pattern depend on the type of acoustic information (RTF, IMS).

## 2. Materials and Methods

In total, 20 patients were recruited who had undergone unilateral hip arthroplasty due to coxarthrosis 8–17 days prior to the intervention. Of those patients, 10 were assigned to the sonification group (SG) and to the control group (CG), respectively. Both groups were parallelized regarding age, duration post-surgery, sex, weight, and height. The specific characteristics of the patients are shown in Table 1.

Every patient was admitted to the same rehabilitation clinic and thus followed a similar rehabilitation program. Patients were excluded if they had multiple artificial joints, implanted pacemakers, a low fitness level (not able to walk for 20 min) or severe pain according to the statement of the patient. Prior to the start of the intervention, patients were informed of the measurements, training process, and intervention procedures, and gave their written consent to participate voluntarily in this study. The study was conducted in accordance with the guidelines stated in the Declaration of Helsinki and the regulations of the Ethical Committee of the Leibniz University Hannover (EV LUH 02/2016).

A clinical test was performed before the intervention started, including a Timed Up and Go test [57] and a functional strength test (getting up from a chair) (Table 1). Hearing ability of the SG was measured using HTTS hearing test software (Version 2.10, SAX GmbH, Berlin, Germany) and cadence was determined during 1 min of walking. At baseline, no significant differences between SG and CG neither for group characteristics nor for clinical tests could be found (*t*-test for independent samples: timed-up and go test *t*(38) = −1.614, *p* = 0.115, *r* = 0.25; functional strength test *t*(18) = 1.659, *p* = 0.115, *r* = 0.36). The intervention sessions were comprised of 10 supervised gait training sessions (TS) with a duration of 20 min each (Figure 1), which were completed within two weeks. TS took place in a 12 m × 15 m gym of the rehabilitation clinic. During gait training, a laptop was placed in the gym to show the patients the temporal progress of the training (Figure 2). Additionally, information concerning number of steps, distance covered, and mean gait speed was given in terms of performance feedback after each TS.

Biweekly, two patients started the gait training intervention. In an admission consultation with a medical specialist of the rehabilitation clinic, patients were informed of the study design and asked to participate in the intervention. To avoid differences in the motivational attitude between the groups, patients were not informed of their group allocation at enrollment, but this information was disclosed postintervention. On average, patients started gait training about two weeks after surgery and they were not able to walk without forearm crutches. Therefore, each patient used forearm crutches in every TS.

For capturing kinematic data, the wireless inertial sensor system MVN Awinda (XSens Technologies B.V., Enschede, the Netherlands) and the software MVN Studio BIOMECH (Version 4.1, XSens Technologies B.V., Enschede, the Netherlands) were used. The sampling rate was set at 60 Hz. Patients of both groups were equipped with the motion analysis system for gait training. Seven inertial measurement units (IMUs) were fixed to the feet, lower legs, upper legs, and pelvis by Velcro straps. Solely SG received the sound of the gait sonification by wireless headphones.

Each TS of 20 min was subdivided into four 5 min blocks consisting of 3 min RTF and 2 min IMS. RTF, providing a low-latency feedback (<100 ms) of the patients’ real gait pattern, was realized by direct data streaming out of the MVN Studio BIOMECH software to Spyder (Version 2.3.5.2., The Scientific Python Development Environment, Spyder Developer Community). In Spyder, an algorithm for detecting touch-down and toe-off of the feet as well as knee extension phase of the right and left leg during gait was applied. The generated kinematic events and periods (ground contact time and knee extension) were synthesized by an implemented Csound (Csound 6, LGPL) module resulting in a succinct sound pattern: The ground contact of the foot can be described in analogy of sound emerging when walking through heavy snow. Knee extension was acoustically represented as a sequence of xylophone strokes, usually a row of 5–7 quickly ascending tones per extension for healthy gait. Consequently, a whole gait cycle resulted in two successive snow compression sounds, each complemented by the xylophone of the contralateral knee extension. To enable a clear mapping between the sound and the according side of the body, the sound of the left leg (knee extension and ground contact of the foot) was four half tones (major third) lower than the sound of the right leg. Further, only the right speaker of the headphone gave the sound of the right leg while the left speaker gave the sound of the left leg.

The same sound pattern was used to generate IMS. Consequently, during IMS mode the patients heard synthesized ‘walking through snow’ sounds and ‘xylophone strokes’ in a fixed tempo, which was chosen based on body height and cadence. More precisely, IMS sounds were pre-recorded based on kinematic data sets to instruct a symmetric gait pattern. RTF and IMS were displayed successively and cumulated in 5 min blocks as it was intended to use enhanced sensorimotor representations formed during RTF for motor planning and execution during IMS. Therefore, exactly the same sound pattern was applied for RTF as well as for IMS. The kinematic data sets to produce a symmetric gait pattern sound were calculated as described below. IMU data of 18 healthy older adults (age: 69.7 ± 7.7 years, height: 1.68 ± 0.1 m, range: 1.53–1.88 m, weight: 72.7 ± 12.7 kg) walking a 10 m distance at three different self-selected gait speeds (‘normal’, ‘slow’, ‘fast’) were collected. A regression analysis was performed considering body height and gait velocity towards several kinematic and spatiotemporal parameters. Resulting regression equations were used to calculate 18 data sets for three different body height ranges with six different cadences, respectively (see Table 2). In this manner, during gait training IMS was chosen based on each patients’ body height and the pre-training cadence. To ensure that IMS acoustically provide a symmetric gait pattern, kinematic data of the right and left leg were shifted by half a gait cycle. The datasets were synthesized and recorded to complete the new gait sonification method.

Recorded gait sequences during RTF (minutes 1–3, 6–8, 11–13, 16–18) and IMS (minutes 4–5, 9–10, 14–15, 19–20) of SG were analyzed separately from each other. To consider the temporal development, TS 1–5 (week 1) and 6–10 (week 2) were clustered. This resulted in four measurement values for each participant, RTF 1–5 (RTF 1), RTF 6–10 (RTF 2), IMS 1–5 (IMS 1), and IMS 6–10 (IMS 2). To ensure comparability and considering fatigue effects, recordings of CG were also divided into RTF parts and IMS parts, even though this group did not receive any sonification. As walking direction during gait training was not specified to the patients, standing phases, direction changes, and turns with a threshold of 10° were excluded from the recordings before final data analysis.

To investigate the effects of the acoustic feedback method on gait symmetry over time (H 1), step lengths of the affected and unaffected leg were considered for week 1 and week 2. Besides, the coefficient of variation (COV) of stride length and COV of stride time were analyzed for RTF 1, RTF 2, IMS 1, and IMS 2 to evaluate gait steadiness and to determine if different effects on the patients’ gait patterns occur depending on different types of acoustic information (H 2). COV was calculated for each patient as the ratio of the standard deviation divided by the mean of stride length (m) respectively stride time (ms). Resulting values were multiplied by a hundred to display COV in percent. Basic parameters as gait speed, cadence, stride length, and stride time were also analyzed, to give a general view on the gait quality of the patients, and to allow for comparability with other findings.

Parameters were calculated using a MATLAB routine (R2016a, The MathWork inc., Natick, MA, USA), in which touch-down and toe-off of the feet were detected by determining peak accelerations of the feet in z-direction (vertical). The interval between touch-down of one foot and touch-down of the contralateral foot was defined as step, and the interval between touch-down of one foot and touch-down of the same foot was defined as stride. Stride time was determined as duration between the touch-down of one foot and the following touch-down of the same foot. Stride length was defined as distance between the touch-down of one foot and the following touch-down of the same foot measured orthogonally to the movement direction.

A three factor (time, mode, and group) repeated measures ANOVA was applied to the dataset using SPSS for Windows 24.0 (Chicago, IL, USA). The significance level was set at 5%. If a significant interaction effect was observed, post-hoc *t*-tests using Holm-Bonferroni sequential correction were performed to identify detailed differences between conditions. Effect sizes (Pearson’s r for *t*-tests and Cohen’s effect size *f* for ANOVA) for the differences between the two groups, time, and mode conditions were calculated to estimate the relevance of any significant difference.

## 3. Results

### 3.1. Symmetry

For the affected leg there were no significant main effects of group (F (1,18) = 0.589, *p* = 0.45, *f* = 0.18) and time (F (1,18) = 0.137, *p* = 0.72, *f* = 0.09). However, a significant group × time interaction could be found (F (1,18) = 6.124, *p* = 0.024, *f* = 0.58). Both groups differed significantly in week 1 with step lengths of 0.58 m ± 0.04 m for SG and 0.54 m ± 0.06 m for CG (t(38)= 2.70, *p* = 0.031, *r* = 0.26). The affected step length of the CG increased (t(19) = −4.45, *p* = 0.001, *r* = 0.44) from week 1 (0.54 m ± 0.06 m) to week 2 (0.57 m ± 0.06 m). In contrast, on a descriptive level, but not significant a decrease from week 1 (0.58 m ± 0.04 m) to week 2 (0.56 m ± 0.07 m) was found for SG (t(1,9) = 1.655, *p* = 0.114, *r* = 0.35).

As a significant increase in step length of the unaffected leg over time could be found for both groups (F (1,18) = 5.70, *p* = 0.028, *f* = 0.56), in SG the step length of both legs converged from week 1 to week 2, while a nearly parallel development of both legs could be observed for subjects of CG (see Figure 3). However, no significant group × side × time interaction was found.

Furthermore, no significant difference between the two modes (RTF, IMS) could be found for the affected and the unaffected step length.

### 3.2. Variability

For the variability of stride length (COV), neither a significant main effect of time (F (1,18) = 2.08, *p* = 0.166, *f* = 0.34) nor a main effect of group (F (1,18) = 2.89, *p* = 0.11, *f* = 0.40) could be found (Table 3). The variability of stride time showed a significant main effect of time (F (1,18) = 7.15, *p* = 0.015, *f* = 0.63), but no significant group effect (F (1,18) = 3.03, *p* = 0.099, *f* = 0.41). Also, there was no group × time interaction neither for the variability of stride length (F (1,18) = 0.50, *p* = 0.488, *f* = 0.17), nor for the variability of stride time (F (1,18) = 2.37, *p* = 0.141, *f* = 0.36).

As the acoustic feedback and the two different modes (RTF and IMS) were only presented to the SG, the statistical analysis of mode effects did not include the CG. For the variability of stride length (Figure 4) a significant difference between the modes RTF and IMS could be found (F (1,9) = 6.50, *p* = 0.03, *f* = 0.85). This is because of a significant difference in stride length variability in the IMS and RTF mode in SG (see Table 3). Moreover, there was a significant mode × time interaction (F (1,9) = 5.63, *p* = 0.042, *f* = 0.79) for the variability of stride length. Considering the variability of stride time (Figure 5) there was no significant mode effect (F (1,9) = 4.13, *p* = 0.073, *f* = 0.68), but solely a significant mode × time interaction (F (1,9) = 7.39, *p* = 0.024, *f* = 0.91).

Both mode × time interaction effects, which were found for the variability of stride length and the variability of stride time, are due to an increase of variability from week 1 to week 2 in RTF. However, post-hoc tests did not reveal any significant effects.

### 3.3. Temporo-Spatial Parameters

The results showed overall significant improvements from week 1 to week 2 in gait speed (F (1,18) = 15.63, *p* = 0.001, *f* = 0.93), cadence (F (1,18) = 20.68, *p* < 0.005, *f* = 1.07), stride time (F (1,18) = 12.9, *p* = 0.002, *f* = 0.85), and stride length (F (1,18) = 6.56, *p* = 0.020, *f* = 0.60) (Table 4). No significant differences between groups were found regarding gait speed, cadence, stride length, and stride time. For SG, a significant mode effect on stride length was found (F (1,9) = 5.199, *p* = 0.049, *f* = 0.76) (RTF mode: 1.16 m ± 0.12 m and IMS mode: 1.14 m ± 0.11 m).

## 4. Discussion

In the present study the effectiveness and feasibility of a new dual mode method of gait sonification for rehabilitation following unilateral hip arthroplasty was investigated. It seems to be crucial that patients following unilateral THA relearn a symmetric gait pattern [58,59]. Therefore, we hypothesized that (H 1) the dual mode acoustic feedback method improves the patients’ gait symmetry over time compared to a control group without acoustic feedback. Results showed converging step lengths of the affected and unaffected leg over time in SG compared to CG. In week 1 a significant higher step length of the affected leg was found for SG compared to CG. Therefore, CG did not show a clear gait asymmetry at the beginning of the intervention. Regarding the temporal development from week 1 to week 2, a nearly parallel increase of step lengths of the affected and unaffected leg could be observed in CG. In contrast in SG, a tendency toward a decrease from week 1 to week 2 could be observed for the affected leg, whereas the step length of the unaffected leg seemed to increase from week 1 to week 2. Statistically, an interaction between SG and CG over time was found for step length of the affected leg. The effect size (*f* = 0.58) indicates a large effect, which suggests that the dual mode method for gait rehabilitation affected gait symmetry. However, this finding needs to be interpreted with caution because the changes observed in step length for patients in the SG did not reach statistical significance and no group × side × time interaction could be found. These statistical results can be due to the following reasons. First, there was an initial difference between groups concerning gait performance. This means that the SG showed a slightly higher gait speed, cadence, stride length, gait variability, and worse gait symmetry, which might have resulted in a higher potential for improvement in CG. Additionally, the used gait sonification method might not yet fully meet the needs of patients following total hip arthroplasty at the time when the intervention started. Finally, it seems that the surgical procedure and the use of crutches limited the patients’ ability to walk freely.

Second, it was hypothesized (H 2), that the effects on the gait pattern depend on the type of acoustic information (RTF, IMS). Within the SG, significantly increased stride lengths were found in RTF training mode compared to IMS mode. Furthermore, in RTF statistically significant higher variability of stride length and stride time were found compared to IMS. Due to these results, H 2 can be accepted. The effect of mode on gait variability could be explained by a reduced stride length variability and stride time variability in IMS mode compared to RTF mode. This might be due to the additional temporal information to guide the patients’ steps during IMS mode and the following elimination of the anticipatory information in RTF mode. Previous investigations have already shown that auditory cueing, which is similar to the IMS mode, improved gait variability of patients with Parkinson’s disease and stroke [60,61,62]. However, studies examining the effects of auditory cueing on healthy older adults showed divergent results [45,63]. For example, Hamacher et al. [63] found higher stride time variability when participants were walking to rhythmic auditory cues compared to normal walking (without auditory cues). Therefore, it appears that in (neurological) healthy older persons auditory feedback addresses different mechanisms and causes different effects on gait compared to patients with Parkinson’s disease or stroke.

In addition, it might be assumed, that in RTF patients tried to calibrate their strides to the previously heard rhythm and tempo in IMS. More precisely, this alteration was likely induced because of the additional sensory information in RTF and the similarity of RTF sound and IMS sound. Potentially, the anticipatory information provided in IMS improved and adjusted the motion concept and motor planning [64]. This effect was used in RTF as here patients’ perception of their own gait pattern was enhanced, which allowed for the comparison between the acoustic symmetric gait (IMS) and the acoustic real gait (RTF). Based on this assumption, an increased internal focus could have occurred in RTF. A resulting tighter and more conscious motor control and a reduced automaticity might have induced an over-correction and consequently the higher stride time and stride length variability in RTF [65,66]. Here, an increased variability should not necessarily be avoided, as during motor relearning processes of trial and error occur, which usually cause noise in the nervous system and in the motor execution [67]. But in later stages of motor relearning, higher automaticity of gait can be achieved. Results also indicate an increase of stride variability (mode × time interaction) in RTF from week 1 to week 2, which seems unexpected, as increased muscle strength and morphologic healing must be assumed over time [68,69]. However, this effect might be a result of increased perceptuomotor control and adaptability [20] and a greater degree of freedom due to less morphological limitations. These factors could have enhanced the conscious motor control and the resulting over-correction, as described above.

The present study investigated a new approach concerning the patient population studied and the acoustic feedback method. The results suggest an impact of the method on the gait pattern in patients with unilateral hip arthroplasty. Therefore, future studies are necessary to investigate the mechanisms (RTF, IMS, or both), that allow to modify selected gait parameter. For example, modifications of the sound and the motion-sound transformation could help to explore the effects of the acoustic feedback method more detailed. Furthermore, the order and the duration of display of RTM and IMS could be changed to investigate whether and to what extent these factors may influence the effectiveness of the method.

Finally, there are some methodological limitations that should be discussed. First, it must be mentioned that in the current study the investigated method was not compared to separate groups receiving only auditory cueing or real-time sonification. However, in this regard more patients would have had to be recruited, which was not possible in a reasonable time. Moreover, it must be considered that the intervention period of two weeks could have been too short to cause unambiguous effects. In particular, regarding the orthopedic limitations of the patients, which usually show a very strong impact on gait [47], a longer intervention period might have resulted in larger effects. Furthermore, the gait training took place in a gym, which allowed the patients to walk freely and consequently with a higher interindividual variability than on a treadmill. However, compared to walking on a treadmill, this setting is closer to real-life conditions of the patients and gait speed as well as gait pattern were not affected by an external device. Using a treadmill would also have increased electromagnetic disturbances, which in general impairs measurement accuracy of IMU systems. Even though electromagnetic disturbances could not be completely avoided in this study, a high quantity of reliable data was analyzed and considered. A further limitation of this investigation is the initial difference between groups concerning gait performance. Both groups were parallelized regarding age, duration post-surgery, sex, weight, and height, but an assessment test to measure initial gait speed, gait variability, and symmetry as well as mobility of the hip and muscle strength was not performed. Though, the duration of the patients’ hospitalization was limited to 18 days, which were structured in terms of a prescribed rehabilitation program and did not allow for additional measurements.

## 5. Conclusions

In the present study, a clear and succinct sound setting was developed and applied, primarily to readjust the patients’ gait symmetry by augmented feedback within a time sensitive perceptual system (RTF) and calibrating the internal symmetry model via IMS. It has been shown that the step lengths of the affected and unaffected leg converged over time in SG compared to a nearly parallel development of both legs in CG. Additionally, in SG a higher variability of stride length and stride time during the RTF training mode compared to the IMS mode was found. The results suggest that the new method is a promising approach, which in future could support gait rehabilitation as well as home-based gait training. Sonification allows for multiple motion sounds and varying motion-sound mappings and therefore, different gait sonification settings and their effects on the gait pattern of orthopedic patients should be examined in future investigations. The introduced method of kinematic gait sonification based on wireless inertial sensors can be easily combined with a variety of already developed motor rehabilitation settings for an enhancement of effectiveness.

## Figures and Tables

**Figure 1 brainsci-09-00066-f001:**
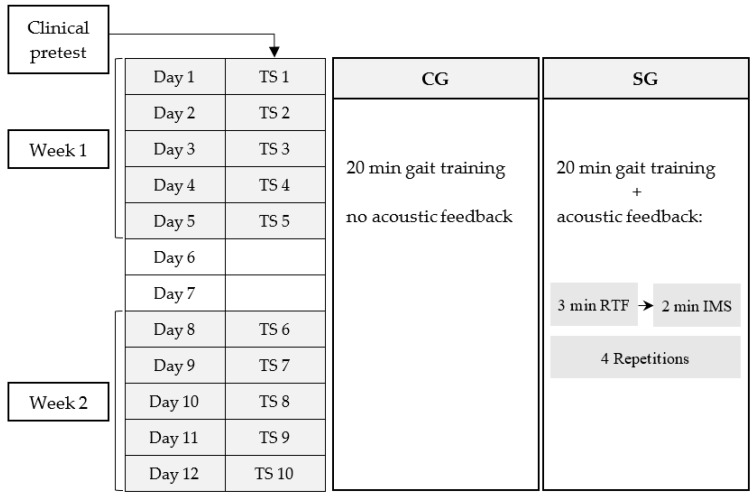
Process of intervention with ten training sessions (TS) on twelve days. The control group (CG) did not receive any acoustic feedback, while the sonification group (SG) received real-time feedback (RTF) alternating with instructive model sequences (IMS).

**Figure 2 brainsci-09-00066-f002:**
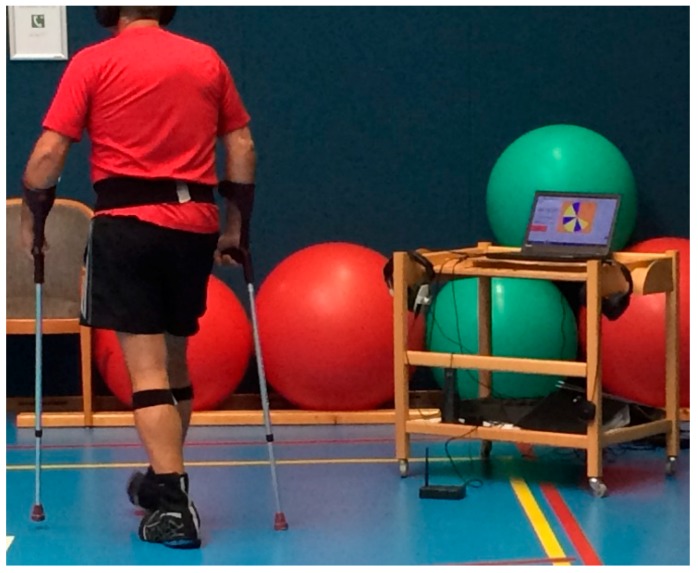
Patient of the sonification group during gait training. The temporal course can be observed on the notebook screen.

**Figure 3 brainsci-09-00066-f003:**
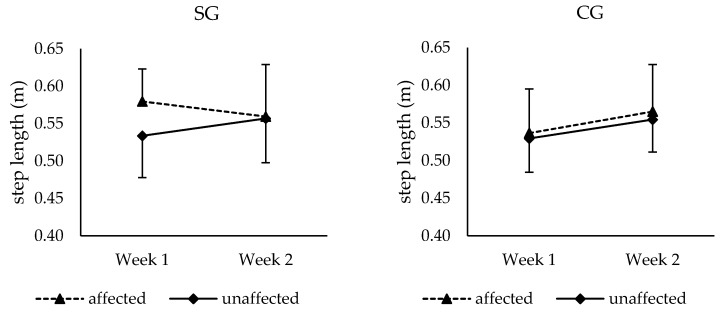
Step length of the affected and unaffected leg for SG (*n* = 10) (left) in week 1 and week 2, step length of the affected and unaffected leg for CG (*n* = 10) (right) in week 1 and week 2. Values are means ± standard deviation.

**Figure 4 brainsci-09-00066-f004:**
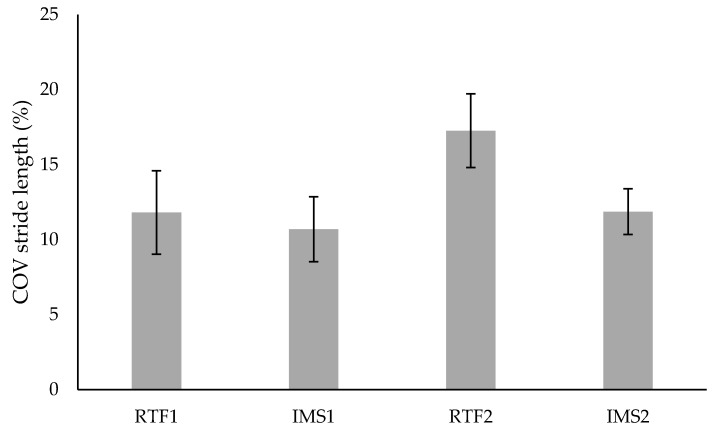
Coefficient of variation (COV) of stride length for SG. Values are means ± standard error.

**Figure 5 brainsci-09-00066-f005:**
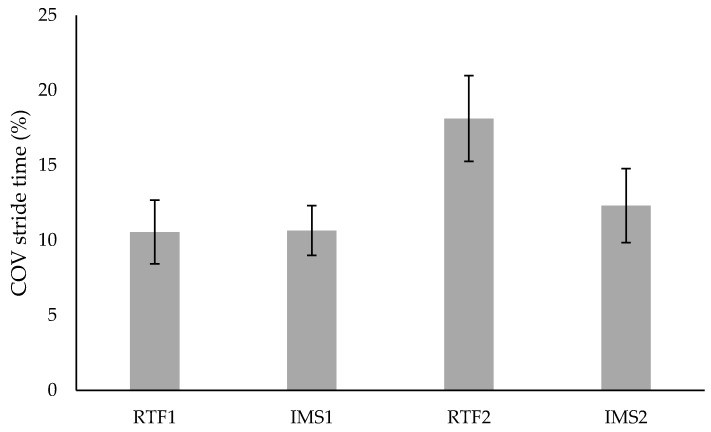
Coefficient of variation (COV) of stride time for SG. Values are means ± standard error.

**Table 1 brainsci-09-00066-t001:** Characteristics of the sonification (SG) and control group (CG). Values are means ± standard deviation.

	SG (*n* = 10)	CG (*n* = 10)
Age (years)	64.0 ± 8.8	61.9 ± 8.4
Sex	9 male, 1 female	7 male, 3 female
Duration post-surgery (days)	11.5 ± 1.6	12.0 ± 2.7
Height (m)	175.1 ± 5.2	176.1 ± 4.1
Weight (kg)	84.4 ± 10.8	85.3 ± 12.4
Timed-up and go (s)	11.58 ± 3.00	13.82 ± 6.24
Functional strength (reps)	12.9 ± 2.2	9.7 ± 5.5

**Table 2 brainsci-09-00066-t002:** Cadences calculated for IMS creation of three different body height ranges and six different gait velocities.

Gait Speed (m × s^−1^)	Cadence (steps × minute^−1^)	Cadence (steps × minute^−1^)	Cadence (steps × minute^−1^)
	155–165 cm	165–175 cm	≥175 cm
0.4	78	69	69
0.6	91	80	79
0.8	102	91	88
1.0	113	101	98
1.2	124	111	106
1.4	135	121	115

**Table 3 brainsci-09-00066-t003:** Coefficient of variation (COV) of stride length and COV of stride time of SG, CG, and in total (SG + CG) for week 1 and week 2. Values are means ± standard deviations.

		COV Stride Length (%)		COV Stride Time (%)	
		RTF	IMS	RTF	IMS
SG (*n* = 10)	Week 1	11.81 ± 8.81	10.69 ± 6.84	10.55 ± 6.73	10.65 ± 5.25
	Week 2	17.26 ± 7.75	11.86 ± 4.81	18.12 ± 9.02	12.32 ±7.81
CG (*n* = 10)	Week 1	8.27 ± 5.55	8.04 ± 6.01	7.90 ± 4.80	7.86 ± 4.69
	Week 2	9.40 ± 5.94	9.17 ± 7.64	9.30 ± 5.87	8.94 ± 6.86
SG + CG (*n* = 20)	Week 1	10.04 ± 7.39	9.36 ± 6.42	9.22 ± 5.85	9.26 ± 5.05
	Week 2	13.33 ± 7.83	10.52 ± 6.36	13.71 ± 8.68	10.63 ± 7.36

**Table 4 brainsci-09-00066-t004:** Gait speed, stride length, cadence, and stride time of SG and CG for week 1 and week 2. Values are means ± standard deviations.

		Gait Speed	Cadence	Stride Length	Stride Time
		(m × s^−1^)	(steps × min^−1^)	(m)	(ms)
SG (*n* = 10)	Week 1	0.93 ± 0.14	99.32 ± 10.76	1.13 ± 0.10	1244.5 ± 136.4
	Week 2	0.97 ± 0.18	103.34 ± 14.56	1.16 ± 0.13	1218.2 ± 161.6
CG (*n* = 10)	Week 1	0.85 ± 0.14	94.88 ± 10.86	1.08 ± 0.10	1290.6 ± 145.1
	Week 2	0.95 ± 0.15	101.80 ± 12.64	1.14 ± 0.09	1218.2 ± 158.7

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
