# Peer review of "Dual Mode Gait Sonification for Rehabilitation After Unilateral Hip Arthroplasty"

_brainsci, 2019, doi:10.3390/brainsci9030066_

Reviewer 1 Report

Thank you for taking my comments into consideration in your revised version of the paper. My previous concerns regarding the justification of the study (motivation); the design choices; and the results of the study and the conclusions that can be derived from them have been mostly addressed. I have only one more concern about the first sentence of the conclusion section: "It has been shown, that the applied dual mode gait sonification method has the potential to improve gait symmetry in patients following unilateral hip arthroplasty."

As far as I understood, this claim couldnt be made as it relates to hypothesis H1 "the dual mode  acoustic feedback method improves the patients’ gait symmetry over time compared to a control group without acoustic feedback.", which as the authors indicate "With regard to the presented results, this hypothesis can neither be rejected nor accepted."

I suggest the authors tone down this sentence. 

Author Response

Dear Reviewer,

we would like to thank you for the great support and the comments, which helped us to improve the manuscript. We have addressed all of them for the revision of the manuscript. You find our response below.

Comments and Suggestions for Authors

Thank you for taking my comments into consideration in your revised version of the paper. My previous concerns regarding the justification of the study (motivation); the design choices; and the results of the study and the conclusions that can be derived from them have been mostly addressed. I have only one more concern about the first sentence of the conclusion section: "It has been shown, that the applied dual mode gait sonification method has the potential to improve gait symmetry in patients following unilateral hip arthroplasty."

As far as I understood, this claim couldn’t be made as it relates to hypothesis H1 "the dual mode acoustic feedback method improves the patients’ gait symmetry over time compared to a control group without acoustic feedback.", which as the authors indicate "With regard to the presented results, this hypothesis can neither be rejected nor accepted."

I suggest the authors tone down this sentence.

Thanks for the advice. We deleted the first sentence in the conclusion section and the mentioned sentence in the discussion ("the dual mode acoustic feedback method improves the patients’ gait symmetry over time compared to a control group without acoustic feedback."). Additionally, we restructured the conclusion to tone down the interpretation of the results.

Reviewer 2 Report

Overall comments:

The authors have done a great job revising the manuscript and the flow of the text has improved considerably. The introduction and methods sections have improved greatly. The Results and Discussion sections still need some work to improve the flow of content and clarify some points.

Editing suggestions:

Line 20: Add: “ … was found during the RTF training mode in comparison to the IMS mode

              Remove “…probably indicating enhanced motor relearning in this mode”.

Line 21: Change “consequently” for “therefore”.

Line 28: Remove comma: “ … fundamental importance for motor learning and re-learning in rehabilitation”.

Line 29: Edit

“An expansion of perception beyond habitual sensorimotor feedback (i.e. augmented feedback)”

Line 33: Remove comma: “preferred as it addresses ….”

Line 34: Change “humans brain” to “human brain”

Line 37: Remove comma “… be considered an alternative, particularly related to cyclic movements, as the human …”

Lines 52-54: The definition of acoustic error feedback is a little vague. Small editing of the sentence would help clarify it. For instance, it says “acoustic error feedback defines certain ranges and compares …”  Certain ranges of what exactly?

Line 54: The definition of “instructing or cueing movements” is also not very clear and a reader not familiar with this field of research would not grasp the concept clearly from what is stated here. Please rephrase.

Line 54: Change “besides” for another preposition such as “additionally” or “on the other hand”, and remove the comma between “instructing or cueing movements”.

Lines 64-66: Use the past tense to describe the Schauer and Mauritz study.

Line 72: It is not clear what “symmetry matched auditory cues affected gait steadiness” means here. Please rephrase to clarify the statement.

Lines 73-74: Remove comma “ … there is no evidence whether auditory cues …”

Line 75: Not just in healthy older persons but also patients, correct?

Line 77: Remove “Even” at the beginning of the sentence.

Line 80: Change “consequently” to “however” at the beginning of the sentence to clarify the connection between the previous and following sentences.

Lines 83, 84, 85: Remove comma: “considered that the prevalence …”; “patients who suffer”; “rehabilitation system that does not require”.

Line 85: Perhaps the word “explicit understanding” is not the most appropriate term here. Maybe “high cognitive effort” or “high attentional cost” would suit best.

Lines 92-93: It needs to be clearer that the main distinction between the real-time feedback model and the instruction model is that, while the auditory feedback in the RTF is triggered by the patient’s movements, the IMS presents the auditory information at a fixed tempo, much like a metronome or a cueing acoustic signal. This needs to be clearly stated here, in the definition paragraph (lines 43-56), and throughout the manuscript.

Line 116: It is not clear what the information “hip arthroplasty lasting over nine months in total” refers to. Does it mean that the hip arthroplasty was done 9 months prior to the study? Or that patients have had the condition for over 9 months? Please clarify.

Line 117: Specify that “over time” here actually means “over 2 weeks”.

Line 121: Change “have” to “had”

Line 148: The name of the Time Up and Go test should be in capital letters and cited.

Podsiadlo, D., & Richardson, S. (1991). The timed “Up & Go”: a test of basic functional mobility for frail elderly persons. Journal of the American geriatrics Society39(2), 142-148.

Lines 149-150: Rephrase: “A t-test for independent samples revealed that there were no significant differences …”

Lines 152-153: The sentence “also, hearing ability …” seems disconnected here. Perhaps, include this information with the sentence at the beginning of the paragraph (line 148) and include the information about the hearing test here.

Lines 156: Change “besides” to “additionally”

Figure 1: The figure is much improved. One suggestion is to have only the arrow connecting the 3 min RTF block and 2 min IMS and place the “4 repetitions” information below. You probably don’t need the arrow connecting the “4 repetitions” to the other blocks.

Lines 174-176: I’d suggest the authors remove or rephrase the sentence:

“Therefore, simultaneously hospitalized patients were assigned to the same group to prevent information transfer between CG and SG.”

This sentence (as it is) just raises some unnecessary concerns about the study allocation.

The authors can probably phrase this as follows:

“ … groups, patients were not informed of their group allocation at enrollment, but this information was disclosed post-intervention”.

Line 182: Start a new sentence “motion analysis system for gait training. Seven …”

Line 185: Change to “subdivided into four 5-minute blocks consisting of …”

Line 194: Were the xylophone tone sequence ascending or descending? Or did in change so that when say the right leg extended there was an ascending tone sequence and when the left leg extended the sound sequence descended? Please include a brief description of the characteristics of these xylophone tones.

Line 198: Change “given rhythm and tempo” for “fixed tempo”.

Line 199: Perhaps this is a clearer way to describe it: “IMS sounds were pre-recorded based on …”

Line 202: Remove comma “healthy older adults”

Line 204: Remove “had been recorded and analyzed”.

Line 208: Change “could be chosen” to “was chosen based on each patient’s body height”.

Line 208: “previously determined cadence” here means the baseline or pre-training cadence?

Lines 211-213: Perhaps these sentences would fit better at the beginning of the paragraph (line 197).

Line 256-269: Please report the main effect of time and the main effect of group before reporting the interaction time*group.

Line 256: Would it be better to state “step length of the affected leg” instead of “affected step length”?

Figure 3, 4, 5: I’m not sure what the guidelines for figure formatting are, but I suggest having all figures with a white background instead of green.

Line 269: Please also mention that there was no interaction between mode and gait symmetry.

Line 273: Start text in a new paragraph (see line 255).

Lines 273-305: The section reporting the results relating to variability is very hard to follow. Please restructure this section.

The authors don’t need to exclude the report on the group analysis entirely. My comment in the first round of revisions was that it didn’t make much sense to report a group*mode or group*mode*time interaction because only one group received the intervention. But it is still relevant to report if there was a main effect and an interaction between group*time because this will tell us if there were changes in variability over 2 weeks for the control group as well as for the sonification group.

So, my suggestion is to divide this section 3.2. as follows:

1st paragraph: report the main effect of group and time for stride length and stride time. If there is a significant interaction between group*time for these variables, please report the comparison analysis to inform which group changed in variability for stride length and stride time and how much. Perhaps it would be better to include a table summarizing all Means and SD.

2nd paragraph: report the main effect of mode and mode*time interaction in stride length for the sonification group.

3rd paragraph: report the main effect of mode and mode*time in stride time for the SG group.

Lines 275-276: “This is because of an increase in stride length variability from IMS to RTF mode in SG (…)”

This statement needs to follow the report of the main effect of mode (line 273-274) to improve the flow and comprehension of the results.

Also, change “increase” to “significant difference” and “from” to “in”: “This is because of a significant difference in stride length variability in the IMS and RTF modes in SG (…)”.

Line 276-277: Move this statement after the mode*time interaction is reported: “an increase in stride length variability over time (…)”.

Line 279: Was there a main effect of mode in stride time? Please report.

Lines 281-283: It looks like the authors are mixing a bit the wording and how to report the results referring to the main effect of time and the interaction between mode*time.

The manner that the results are reported here compares stride time for the IMS and the RTF modes in relation to TIME. Therefore, it is not appropriate to use words such as “change” and “increase” as we would be interested in “differences”.

So, the better way to state the results regarding the main effect of time would be: “in week 1 there were no significant differences in stride time variability between the IMS and RTF modes for SG”.  Consequently, the next information should be if there were differences between the IMS and RTF modes in week 2.

If the authors want to report the results for the mode*time interaction and whether there were changes in stride time from week 1 to week 2 for both modes, the authors should report Means and SD for Week 1 and Week 2 side by side for each mode, so that the reader can understand where there were changes or increases.

For example: “in week 2, there was a significant increase in stride time for the IMS mode (Mean/SD week 1 for IMS, Mean/SD week 2 for IMS) as well as for the RTF mode (Mean/SD week 1 for RTF, Mean/SD week 2 for RTF)”. The results relating to week 1 should also be reported in the same manner.

Line 308: Change “p = 0.000” to “p < 0.005”

Line 309: Delete “were measured for both groups”.

Line 311: p = 0.49; Did you mean p = 0.049?

Line 325: The content of the discussion and the interpretation of the results has improved considerably but the flow of the text needs to be improved. Perhaps the authors could focus the first paragraph(s) of the discussion on the results relating to gait symmetry, and then follow with the discussion of the results on gait steadiness/variability, changes in gait parameters (speed, cadence, etc.) for both groups, and finally study limitations.

For example, move the paragraph from lines 342-355 directly after the first sentence (line 327).

Line 345: After stating the hypothesis, report the study results: “Results showed converging step lengths of the affected and unaffected leg over time in SG compared with a parallel development of both legs in CG.”

Lines 345-346: Delete: “With regard to the presented results, this hypothesis can neither be rejected nor accepted.”

Line 347: Add: “was found for CG compared to SG”.

Line 348: Delete “However”

Line 349: The authors need to revise this sentence.

The step length of the affected leg showed a tendency toward a DECREASE from week 1 to week 2, whereas the step length of the unaffected leg seemed to INCREASE from week 1 to week 2.  

Line 350: Delete comma “This suggests that …”; and change “affects” to “affected”.

Line 350-351: Please add a sentence such as: “However, this finding needs to be interpreted with caution because the changes observed in step length for patients in the sonification group did not reach statistical significance”.

Line 353: the acronym THA was not used in the text before. Please define it.

Line 354: Delete “and results indicate”. The study did not directly test whether using crutches or not interfere with the rehabilitation protocol.

Line 357: State the results after the hypothesis. “Within the SG, higher variability of stride length and stride time was found during the RTF training mode than in the IMS mode”.

Line 358-362: If I understood correctly, the authors argued here that the increase in variability in the RTF is because patients tried to match their strides to the IMS tempo.

I’m not sure this interpretation for the increase in variability in the RTF due to IMS can be sustained.

There are several studies suggesting the opposite, that presenting an auditory cue at a fixed tempo primes the motor system and DECREASES gait variability even after the cues are removed (McIntosh et al., 1997, 1998). So, based on the literature, it is more possible that removing the anticipatory information provided by the IMS increased stride variability in the RTF mode because patients no longer had the temporal information to guide their movement. 

If the authors would like to sustain the argument that the increase in variability is because patients tried to match their strides to the IMS tempo, please provide studies evidencing that IMS (or auditory cues at a fixed tempo) would indeed INCREASE the variability of a motor task conducted after the cues are removed.

McIntosh, G. C., Brown, S. H., Rice, R. R., and Thaut, M. H. (1997). Rhythmic

auditory-motor facilitation of gait patterns in patients with Parkinson’s disease.

J. Neurol. Neurosurg. Psychiatry 62, 22–26. doi: 10.1136/jnnp.62.1.22

McIntosh, G. C., Rice, R. R., Hurt, C. P., and Thaut, M. H. (1998). Long-term

training effects of rhythmic auditory stimulation on gait in patients with

Parkinson’s disease. Mov. Disord. 13:212.

Lines 364-366: If this statement refers to another possibility (physiological factors) for the interpretation of the results, please remove the preposition “but” at the beginning of the sentence.

Lines 367-369: Are there studies showing that patients would improve muscle strength and other aspects of movements after only 2 weeks of surgery? Please back up your argument with references.

Line 369-375: It is not clear to me how increased in variability during RTF may be linked to increased perceptuomotor control, greater degree of freedom and movement automaticity. Please expand and clarify this argument if the authors decide to maintain these points in the discussion.

Line 376: I’d suggest the authors move the paragraph discussing changes in gait parameters (speed, cadence, etc.) for both groups here, after discussing the results relating to the second hypothesis and following the presentation of the findings in the Results section. This data is another important point to be considered in the interpretation of the study findings.

Line 376: Delete the word “very” in ‘a new approach’.

Also, it may be better to say: “concerning the patient population studied” instead of “the chosen patient group”.

Line 377: Change ‘confirms’ to ‘suggests’.

Line 381: Edit to “the acoustic feedback method”.

Line 382: Remove comma “to investigate whether …”;

  Change to “these factors may influence the effectiveness …”

Line 383, 384: Remove comma “methodological limitation that should …”; “mentioned that in the current study …”

Line 384-385: Did you mean that the study did not compare the combined dual mode with separate groups receiving only auditory cueing or real-time sonification. Please rephrase accordingly.

Line 386: Remove comma “considered that”

Line 395: Remove the word “too”

Line 397: Change “between SG and CG” to “between groups”.

I’d suggest the authors include this limitation regarding group differences at baseline when discussing changes in stride symmetry earlier in the discussion. This point is important for the interpretation of the results, not just a study limitation.

Lines 408-422: I’d suggest the authors leave only the second paragraph (lines 414-422) and delete the sentences from lines 408-413. The summary of results could be added at line 416 (“The results suggest that …”) much like it was done in the abstract with a very brief sentence overviewing the main results.

Author Response

Dear Reviewer,

we would like to thank you for the great support and the comments, which helped us to improve the manuscript. We have addressed all of them for the revision. You will find our responses below.

Overall comments:

The authors have done a great job revising the manuscript and the flow of the text has improved considerably. The introduction and methods sections have improved greatly. The Results and Discussion sections still need some work to improve the flow of content and clarify some points.

Editing suggestions:

1.     Line 20: Add: “ … was found during the RTF training mode in comparison to the IMS mode

Remove “…probably indicating enhanced motor relearning in this mode”.

We rephrased the sentence accordingly.

2.     Line 21: Change “consequently” for “therefore”.

We exchanged the words.

3.     Line 28: Remove comma: “ … fundamental importance for motor learning and re-learning in rehabilitation”.

The comma was removed. This applies also for the comments No. 5, 7, 13, 17, 58, 71, 72 and 74. Thank you for spell checking.

4.     Line 29: Edit

“An expansion of perception beyond habitual sensorimotor feedback (i.e. augmented feedback)”

We changed “augmenting” for “augmented feedback”.

5.     Line 33: Remove comma: “preferred as it addresses ….”

Please see comment No. 3.

6.     Line 34: Change “humans brain” to “human brain”

The error has been corrected.

7.     Line 37: Remove comma “… be considered an alternative, particularly related to cyclic movements, as the human …”

Please see comment No. 3.

8.     Lines 52-54: The definition of acoustic error feedback is a little vague. Small editing of the sentence would help clarify it. For instance, it says “acoustic error feedback defines certain ranges and compares …”  Certain ranges of what exactly?

Thanks for the advice. We edited this definition.

9.     Line 54: The definition of “instructing or cueing movements” is also not very clear and a reader not familiar with this field of research would not grasp the concept clearly from what is stated here. Please rephrase.

Also, this definition was rephrased.

10.  Line 54: Change “besides” for another preposition such as “additionally” or “on the other hand”, and remove the comma between “instructing or cueing movements”.

We removed the comma and changed “besides” for “additionally”.

11.  Lines 64-66: Use the past tense to describe the Schauer and Mauritz study.

The description was corrected accordingly.

12.  Line 72: It is not clear what “symmetry matched auditory cues affected gait steadiness” means here. Please rephrase to clarify the statement.

We added additional information.

13.  Lines 73-74: Remove comma “ … there is no evidence whether auditory cues …”

Please see comment No. 3.

14.  Line 75: Not just in healthy older persons but also patients, correct?

We added “…and orthopedic patients.”

15.  Line 77: Remove “Even” at the beginning of the sentence.

We included this change in the manuscript.

16.  Line 80: Change “consequently” to “however” at the beginning of the sentence to clarify the connection between the previous and following sentences.

“Consequently” and “however” were exchanged.

17.  Lines 83, 84, 85: Remove comma: “considered that the prevalence …”; “patients who suffer”; “rehabilitation system that does not require”.

Please see comment 3.

18.  Line 85: Perhaps the word “explicit understanding” is not the most appropriate term here. Maybe “high cognitive effort” or “high attentional cost” would suit best.

We changed “explicit understanding” to “high attentional cost”.

19.  Lines 92-93: It needs to be clearer that the main distinction between the real-time feedback model and the instruction model is that, while the auditory feedback in the RTF is triggered by the patient’s movements, the IMS presents the auditory information at a fixed tempo, much like a metronome or a cueing acoustic signal. This needs to be clearly stated here, in the definition paragraph (lines 43-56), and throughout the manuscript.

We tried to clarify the distinction between the two modes (RTF and IMS) in the manuscript.

20.  Line 116: It is not clear what the information “hip arthroplasty lasting over nine months in total” refers to. Does it mean that the hip arthroplasty was done 9 months prior to the study? Or that patients have had the condition for over 9 months? Please clarify.

We removed this information as it can be confusing in this context.

21.  Line 117: Specify that “over time” here actually means “over 2 weeks”.

The statement was specified.

22.  Line 121: Change “have” to “had”

We changed the words.

23.  Line 148: The name of the Time Up and Go test should be in capital letters and cited.

Podsiadlo, D., & Richardson, S. (1991). The timed “Up & Go”: a test of basic functional mobility for frail elderly persons. Journal of the American geriatrics Society39(2), 142-148.

Thanks for this information. The spelling was corrected, and the reference was added.

24.  Lines 149-150: Rephrase: “A t-test for independent samples revealed that there were no significant differences …”

The sentence was rephrased.

25.  Lines 152-153: The sentence “also, hearing ability …” seems disconnected here. Perhaps, include this information with the sentence at the beginning of the paragraph (line 148) and include the information about the hearing test here.

Thanks for the suggestion. We moved the sentence to the beginning of the paragraph and added the information about the hearing test.

26.  Lines 156: Change “besides” to “additionally”

The words were changed.

27.  Figure 1: The figure is much improved. One suggestion is to have only the arrow connecting the 3 min RTF block and 2 min IMS and place the “4 repetitions” information below. You probably don’t need the arrow connecting the “4 repetitions” to the other blocks.

Thanks, we modified the figure accordingly.

28.  Lines 174-176: I’d suggest the authors remove or rephrase the sentence:

“Therefore, simultaneously hospitalized patients were assigned to the same group to prevent information transfer between CG and SG.”

This sentence (as it is) just raises some unnecessary concerns about the study allocation.

The authors can probably phrase this as follows:

“ … groups, patients were not informed of their group allocation at enrollment, but this information was disclosed post-intervention”.

This is a very helpful comment. We removed the sentence and instead included the mentioned information.

29.  Line 182: Start a new sentence “motion analysis system for gait training. Seven …”

Thanks for the hint. We started a new sentence.

30.  Line 185: Change to “subdivided into four 5-minute blocks consisting of …”

The sentence was changed accordingly.

31.  Line 194: Were the xylophone tone sequence ascending or descending? Or did it change so that when say the right leg extended there was an ascending tone sequence and when the left leg extended the sound sequence descended? Please include a brief description of the characteristics of these xylophone tones.

We included a brief description with regard to this suggestion.

32.  Line 198: Change “given rhythm and tempo” for “fixed tempo”.

The term was changed.

33.  Line 199: Perhaps this is a clearer way to describe it: “IMS sounds were pre-recorded based on …”

We included this description.

34.  Line 202: Remove comma “healthy older adults”

Please see comment No. 3.

35.  Line 204: Remove “had been recorded and analyzed”.

The term was removed.

36.  Line 208: Change “could be chosen” to “was chosen based on each patient’s body height”.

We changed the wording accordingly.

37.  Line 208: “previously determined cadence” here means the baseline or pre-training cadence?

“Previously determined cadence” means the pre-training cadence. We included this information.

38.  Lines 211-213: Perhaps these sentences would fit better at the beginning of the paragraph.

The sentences were moved to the beginning of the paragraph. “More precisely, IMS sounds were pre-recorded based on kinematic data sets to instruct a symmetric gait pattern. RTF and IMS were displayed successively and cumulated in 5 min blocks as it was intended to use enhanced…”

39.  Line 256-269: Please report the main effect of time and the main effect of group before reporting the interaction time*group.

We included this information at the beginning of the results section.

40.  Line 256: Would it be better to state “step length of the affected leg” instead of “affected step length”?

Thanks for the note. We changed the wording.

41.  Figure 3, 4, 5: I’m not sure what the guidelines for figure formatting are, but I suggest having all figures with a white background instead of green.

Thanks for the note. In our document all figures have a white background. We don’t know the reason for this discrepancy, maybe it is due to the “Track Change” function.

42.  Line 269: Please also mention that there was no interaction between mode and gait symmetry.

We included this information.

43.  Line 273: Start text in a new paragraph (see line 255).

We changed the formatting.

44.  Lines 273-305: The section reporting the results relating to variability is very hard to follow. Please restructure this section.

The authors don’t need to exclude the report on the group analysis entirely. My comment in the first round of revisions was that it didn’t make much sense to report a group*mode or group*mode*time interaction because only one group received the intervention. But it is still relevant to report if there was a main effect and an interaction between group*time because this will tell us if there were changes in variability over 2 weeks for the control group as well as for the sonification group.

So, my suggestion is to divide this section 3.2. as follows:

1st paragraph: report the main effect of group and time for stride length and stride time. If there is a significant interaction between group*time for these variables, please report the comparison analysis to inform which group changed in variability for stride length and stride time and how much. Perhaps it would be better to include a table summarizing all Means and SD.

2nd paragraph: report the main effect of mode and mode*time interaction in stride length for the sonification group.

3rd paragraph: report the main effect of mode and mode*time in stride time for the SG group.

Thank you very much for the support. We restructured this section according to your suggestion. Also, a table (Table 3) summarizing the stride length and stride time variability was included.

45.  Lines 275-276: “This is because of an increase in stride length variability from IMS to RTF mode in SG (…)”

This statement needs to follow the report of the main effect of mode (line 273-274) to improve the flow and comprehension of the results.

Also, change “increase” to “significant difference” and “from” to “in”: “This is because of a significant difference in stride length variability in the IMS and RTF modes in SG (…)”.

The sentence was moved behind the report of the main effect of mode and rephrased accordingly (“This is because of a significant difference in stride length variability in the IMS and RTF mode in SG (s. Table 3).”).

46.  Line 276-277: Move this statement after the mode*time interaction is reported: “an increase in stride length variability over time (…)”.

This statement is now following the report of the mode*time interaction.

47.  Line 279: Was there a main effect of mode in stride time? Please report.

Thanks for the note. There was no main effect of mode in stride time (F(1,9) = 4.13, p = 0.073, f = 0.68). We included this information in the text.

48.  Lines 281-283: It looks like the authors are mixing a bit the wording and how to report the results referring to the main effect of time and the interaction between mode*time.

The manner that the results are reported here compares stride time for the IMS and the RTF modes in relation to TIME. Therefore, it is not appropriate to use words such as “change” and “increase” as we would be interested in “differences”.

So, the better way to state the results regarding the main effect of time would be: “in week 1 there were no significant differences in stride time variability between the IMS and RTF modes for SG”.  Consequently, the next information should be if there were differences between the IMS and RTF modes in week 2.

If the authors want to report the results for the mode*time interaction and whether there were changes in stride time from week 1 to week 2 for both modes, the authors should report Means and SD for Week 1 and Week 2 side by side for each mode, so that the reader can understand where there were changes or increases.

For example: “in week 2, there was a significant increase in stride time for the IMS mode (Mean/SD week 1 for IMS, Mean/SD week 2 for IMS) as well as for the RTF mode (Mean/SD week 1 for RTF, Mean/SD week 2 for RTF)”. The results relating to week 1 should also be reported in the same manner.

Indeed, the wording is misleading. We restructured and rephrased this paragraph and hope the description is clarified now. Also, Mean and SD are now shown in the table to present the differences.

49.  Line 308: Change “p = 0.000” to “p < 0.005”

The information was changed.

50.  Line 309: Delete “were measured for both groups”.

We deleted the mentioned statement.

51.  Line 311: p = 0.49; Did you mean p = 0.049?

Thank you for the correction. This was a typing error, p = 0.049.

52.  Line 325: The content of the discussion and the interpretation of the results has improved considerably but the flow of the text needs to be improved. Perhaps the authors could focus the first paragraph(s) of the discussion on the results relating to gait symmetry, and then follow with the discussion of the results on gait steadiness/variability, changes in gait parameters (speed, cadence, etc.) for both groups, and finally study limitations.

For example, move the paragraph from lines 342-355 directly after the first sentence (line 327).

Thanks for the helpful suggestion. We restructured the discussion, beginning with gait symmetry (hypothesis 1), gait variability (hypothesis 2) and followed by the general changes in gait parameters and the study limitations.

53.  Line 345: After stating the hypothesis, report the study results: “Results showed converging step lengths of the affected and unaffected leg over time in SG compared with a parallel development of both legs in CG.”

The study results were moved after the statement concerning the hypothesis.

54.  Lines 345-346: Delete: “With regard to the presented results, this hypothesis can neither be rejected nor accepted.”

We deleted this sentence.

55.  Line 347: Add: “was found for CG compared to SG”.

The information was added.

56.  Line 348: Delete “However”

The word was deleted.

57.  Line 349: The authors need to revise this sentence.

The step length of the affected leg showed a tendency toward a DECREASE from week 1 to week 2, whereas the step length of the unaffected leg seemed to INCREASE from week 1 to week 2.

Thanks for the attentive reading. We rephrased this sentence.

58.  Line 350: Delete comma “This suggests that …”; and change “affects” to “affected”.

Please see comment No. 3.

59.  Line 350-351: Please add a sentence such as: “However, this finding needs to be interpreted with caution because the changes observed in step length for patients in the sonification group did not reach statistical significance”.

We added the mentioned information by modifying the previous sentence.

60.  Line 353: the acronym THA was not used in the text before. Please define it.

The acronym THA was now defined in the introduction.

61.  Line 354: Delete “and results indicate”. The study did not directly test whether using crutches or not interfere with the rehabilitation protocol.

We deleted this wording.

62.  Line 357: State the results after the hypothesis. “Within the SG, higher variability of stride length and stride time was found during the RTF training mode than in the IMS mode”.

The results were moved behind the hypothesis (“Within the SG, significantly increased stride lengths were found in RTF training mode compared to IMS mode. Furthermore, in RTF…”).

63.  Line 358-362: If I understood correctly, the authors argued here that the increase in variability in the RTF is because patients tried to match their strides to the IMS tempo.

I’m not sure this interpretation for the increase in variability in the RTF due to IMS can be sustained.

There are several studies suggesting the opposite, that presenting an auditory cue at a fixed tempo primes the motor system and DECREASES gait variability even after the cues are removed (McIntosh et al., 1997, 1998). So, based on the literature, it is more possible that removing the anticipatory information provided by the IMS increased stride variability in the RTF mode because patients no longer had the temporal information to guide their movement.  

If the authors would like to sustain the argument that the increase in variability is because patients tried to match their strides to the IMS tempo, please provide studies evidencing that IMS (or auditory cues at a fixed tempo) would indeed INCREASE the variability of a motor task conducted after the cues are removed.

McIntosh, G. C., Brown, S. H., Rice, R. R., and Thaut, M. H. (1997). Rhythmic auditory-motor facilitation of gait patterns in patients with Parkinson’s disease. J. Neurol. Neurosurg. Psychiatry 62, 22–26. doi: 10.1136/jnnp.62.1.22

McIntosh, G. C., Rice, R. R., Hurt, C. P., and Thaut, M. H. (1998). Long-term training effects of rhythmic auditory stimulation on gait in patients with Parkinson’s disease. Mov. Disord. 13:212.

Thank you for the critical reflection on this point. We included the possibility that the difference between IMS and RTF is due to a reduced variability in IMS. However, we would also like to retain our previous assumption, as the listed (McIntosh et al., 1997 & McIntish et al., 1998) and further studies showing that auditory cues decrease gait variability focus on neurologically impaired patients. There are studies on healthy older adults showing divergent effects of auditory cueing (Hamacher et al., 2016; Brodie et al., 2015).

We could not find any studies evidencing that IMS increases the variability of a motor task conducted after the cues are removed, but the interpretation of our results is not based on this assumption. We rather stated that the RTF mode causes the increase in variability and we try to clarify this assumption and sustain it by additional references in the revised discussion section.

Hamacher, D.; Hamacher, D.; Herold, F.; Schega, L. Effect of dual tasks on gait variability in walking to auditory cues in older and young individuals. Experimental brain research 2016, 234, 3555–3563, doi:10.1007/s00221-016-4754-x.

Brodie, M.A.D.; Dean, R.T.; Beijer, T.R.; Canning, C.G.; Smith, S.T.; Menant, J.C.; Lord, S.R. Symmetry matched auditory cues improve gait steadiness in most people with Parkinson's disease but not in healthy older people. Journal of Parkinson's disease 2015, 5, 105–116, doi:10.3233/JPD-140430.

64.  Lines 364-366: If this statement refers to another possibility (physiological factors) for the interpretation of the results, please remove the preposition “but” at the beginning of the sentence.

The word was removed.

65.  Lines 367-369: Are there studies showing that patients would improve muscle strength and other aspects of movements after only 2 weeks of surgery? Please back up your argument with references.

We included the following reference.

Winther, S.B.; Husby, V.S.; Foss, O.A.; Wik, T.S.; Svenningsen, S.; Engdal, M.; Haugan, K.; Husby, O.S. Muscular strength after total hip arthroplasty. A prospective comparison of 3 surgical approaches. Acta orthopaedica 2016, 87, 22–28, doi:10.3109/17453674.2015.1068032.

66.  Line 369-375: It is not clear to me how increased in variability during RTF may be linked to increased perceptuomotor control, greater degree of freedom and movement automaticity. Please expand and clarify this argument if the authors decide to maintain these points in the discussion.

We explained this argument in the previous assumption and tried to clarify the link.

67.  Line 376: I’d suggest the authors move the paragraph discussing changes in gait parameters (speed, cadence, etc.) for both groups here, after discussing the results relating to the second hypothesis and following the presentation of the findings in the Results section. This data is another important point to be considered in the interpretation of the study findings.

The mentioned paragraph was moved after discussing the second hypothesis.

68.  Line 376: Delete the word “very” in ‘a new approach’.

Also, it may be better to say: “concerning the patient population studied” instead of “the chosen patient group”.

We changed the wording accordingly.

69.  Line 377: Change ‘confirms’ to ‘suggests’.

We exchanged the two words.

70.  Line 381: Edit to “the acoustic feedback method”.

The sentence was edited accordingly.

71.  Line 382: Remove comma “to investigate whether …”;

Change to “these factors may influence the effectiveness …”

The comma was removed, and the wording was changed.

72.  Line 383, 384: Remove comma “methodological limitation that should …”; “mentioned that in the current study …”

Please see comment No. 3.

73.  Line 384-385: Did you mean that the study did not compare the combined dual mode with separate groups receiving only auditory cueing or real-time sonification. Please rephrase accordingly.

We rephrased the sentence.

74.  Line 386: Remove comma “considered that”

Please see comment No. 3.

75.  Line 395: Remove the word “too”

The word was removed.

76.  Line 397: Change “between SG and CG” to “between groups”.

I’d suggest the authors include this limitation regarding group differences at baseline when discussing changes in stride symmetry earlier in the discussion. This point is important for the interpretation of the results, not just a study limitation.

We rephrased the sentence accordingly and included this information when discussing gait symmetry.

77.  Lines 408-422: I’d suggest the authors leave only the second paragraph (lines 414-422) and delete the sentences from lines 408-413. The summary of results could be added at line 416 (“The results suggest that …”) much like it was done in the abstract with a very brief sentence overviewing the main results.

We restructured the conclusion section accordingly.

This manuscript is a resubmission of an earlier submission. The following is a list of the peer review reports and author responses from that submission.

Round  1

Reviewer 1 Report
Summary:

In this study, the authors propose a new gait sonification method with two different modes (real-time feedback (RTF), and instructive model sequences (IMS)) for gait rehabilitation of patients with unilateral Hip Arthroplasty. The impact of using this system on gait symmetry and steadiness was investigated on two groups of 10 patients each (sonification and control groups). Results showed converging step lengths of the affected and unaffected leg over time in the sonification group, and higher variability of one of the modes vs the other. The authors discuss these results and their potential to support gait rehabilitation.

Evaluation:

The possibility to use sonification of movementhas great potential to improve motor rehabilitation; understanding its effects and how to optimize them is a relevant topic. Studies into the design of such sonification and training are a necessary step in order to inform future design processes and investigations. This paper offers some interesting ideas in this direction.

The paper’s organization, style and the written presentation are overall good. The introduction sets well the topic of the paper, and provides a good summary of the related work and references to previous related works. However, I have three major concerns about the paper that I hope the authors can address in their revision. These relate to the justification of the study (motivation); the design choices; and the results of the study and the conclusions that can be derived from them, as I explain below:

1) Study motivation: the need to provide a better motivation for the study - why a new approach is needed instead of adopting an approach already tested in other populations?;

2) Design choices: Provide a better justification of the design decisions:  why was that specific combination of RTF and IMS chosen? And the sounds? What was the reasoning behind of displaying the progress to participants, and what were the effects of doing it?;

3) Results/conclusions : with regards to symmetry, the decrease on asymmetry of step time in SG was not significant, and there was no significant group*side interaction for step length. For the temporo-Spatial parameters there were no significant differences between groups. There was an increase in variability for SG but it is not clear that this is something good. As such, it is difficult to make any conclusions based on the reported results. The authors should better explain their contribution and tone down the claims based on non-significant results.

Below I provide a few more detailed comments that I hope help the authors to improve their contribution.

DETAILED COMMENTS:

***Abstract: 

-“à 20 minutes” – perhaps change “á” for “lasting”

-"...compared with a nearly parallel development in CG. " this sentence is not very clear until one reads the paper - please rephrase. Perhaps  "...compared with a nearly parallel development of both legs in CG. "

-"... probably indicating enhanced motor relearning " - again, here is not clear whether the enhanced motor relearning was in RTF or IMS. is variability good in this context?

****Introduction

-line 47: "instructing, or cueing movements means, that specific sound characteristics are predetermined, e.g. rhythm, rate or amplitude, with the aim of affecting a particular movement. " - this sentence is not clear, because the same applies to realtime feedback and acoustic error feedback - could you please clarify?

-line 54: it would be good to keep in this paragraph the same nomenclature introduced in the previous paragraph. "applied musical motor feedback " - is this real-time feedback?

-line 68: check this additional reference on the effects of a different type of auditory feedback on healthy persons’ gait pattern: Tajadura-Jimenez et al., 2015, ACM CHI. The same system was used in a pilot study for rehabilitation of people with CRPS - Tajadura-Jimenez et al., 2017. Frontiers in Neuroscience

-last paragraph (starting in line 75) - i felt that here a better motivation/rationale for the specific design of the study is missing. why was this combination chosen? why a new approach is needed instead of adopting an approach already tested in other populations?

****Methods

methods: 

-line 91: "Both groups were parallelized regarding age, duration post-surgery,sex, weight, and height. " - did you check for potential statistic differences? 

-lines 104-105: what was the aim of showing the patients the temporal progress of the training? was there any effect resulting from it?

-lines 137-138: why were the sounds produced "when walking through heavy snow" and "xylophone strokes " chosen?

***Results

-Symmetry: 

**please make clear at the beginning of this section what factors were significant in the ANOVA. as far as i understand, there was an effect of group, and an effect of time. was there also a significant group*time interaction - which would then be followed by the t-tests reported?

-the finding is due to a bigger step length for the affected leg in TS 1-5, which disappears in TS 6-10. there is no asymmetry in the CG group.

Figure 3: to improve readability of the figure, it would be good to add the names of the groups to the graph itself, rather than just keeping it in the legend

Reviewer 2 Report

Brief summary:

The study investigated the feasibility and effectiveness of an intervention combining real-time feedback and movement cueing to improve gait symmetry and steadiness after hip arthroplasty. Findings revealed different gait patterns after 2 weeks for patients in the sonification group in relation to patients in a control condition, suggesting that gait rehabilitation with auditory feedback changes the pattern of movement recovery. Secondly, the results indicated important differences in relation to the sonification mode used as higher stride variability were observed during real-time sonification in comparison to auditory signals presented at a fixed temporal cue.

Overall comments

The study presents findings that are of great interest to the scientific community, however, there are some central issues that need to be addressed.

Primarily, the interpretation of the study findings needs to be carefully revisited in the discussion and conclusion. The interpretation of the results seems strongly biased toward sonification as the results of the control group are not well discussed and not fully considered. For instance, the study found that there were significant changes in step length after 2 weeks for patients in the control group, but these changes were somewhat different from those observed for the patients in the sonification group. Also, temporo-spatial outcomes (gait speed, cadence, stride length, stride time) increased significantly for both groups.

I recommend the authors to completely restructure and shorten the discussion section, and focus their discussion points on the two main findings (which are clearly pointed in the abstract).

Specific comments:

Introduction:

Lines 31-32: Please include references.

“Different sensory feedback systems have been developed for such purposes and have already been used in sports and rehabilitation (REFERENCES).”

Line 32: What does “easy to follow” mean in this context?

Lines 32-34: In this sentence, the authors make a general statement that implies that the application of visual feedback is limited because it is involved in environmental perception.

Please rephrase this sentence to make it clear to the reader that this statement refers to movement training/rehabilitation context. 

Also, please include references supporting this statement.

Lines 37-39: The examples of cyclic movements presented here are mostly sport-related (running, rowing, cycling, or swimming). Given that the paper focuses on rehabilitation, perhaps it is more appropriate to include more general (cyclic) movements such as gait, arm reaching, etc.

Lines 41-49: Please include references supporting the statements in this paragraph.

While some studies used real-time feedback, others have shown effects of acoustic error feedback or instructing or cueing movements (REFERENCES). In this respect, a distinction can be made between real-time feedback, acoustic error feedback, and instructing or cueing movements: Real-time feedback reflects a movement acoustically, which means kinetic or kinematic data are measured and assigned to a specific sound with low latency (REFERENCE). Compared with this, acoustic error feedback defines certain ranges and compares measured motion data to these (also labeled as ‘bandwidth-feedback’) [17]. A sound signal is only played, if a given range is crossed. Besides, instructing, or cueing movements means, that specific sound characteristics are predetermined, e.g. rhythm, rate or amplitude, with the aim of affecting a particular movement (REFERENCE).

Lines 42-45: This sentence is too long. Perhaps break the sentence before defining ‘real-time feedback’ (Line 43).

Lines 52-54: It might be relevant to briefly present the study results for Rodger et al. and Schauer and Mauritz.

Lines 55-56: The title of reference 23 is missing from the Reference List.

Line 57: Remove ‘though’ at the beginning of the sentence.

Lines 63-64: Please include a reference.

“Usually gait rehabilitation after hip or knee arthroplasty is associated with a large effort of time and personnel (REF.)”

Lines 68-74: In this paragraph, the authors introduce study results in healthy individuals after they presented the study problem and the motivation for the study at the end of the previous paragraph. This breaks the flow between paragraphs in the introduction.

I suggest the authors move the information within this paragraph before addressing studies on patient populations (example: after the first sentence of the previous paragraph – line 51).

Line 76: Please include a brief description of the sonification approach used in the study, describing what exactly was the kinematic real-time feedback and instructive model sequences used. Also, include the reasoning for using a dual mode method instead of testing these approaches in separate/independent groups.

Lines 85-87: Do these sentences need to be in bold text?

Materials and Methods

Line 90: Change ‘ago’ for ‘prior’.

Line 95: How was pain assessed for exclusion?

Line 101: Was there a significant difference between groups at baseline, particularly for Time Up and Go and functional strength?

Line 101: How was hearing ability assessed?

Line 103: Please include that the intervention sessions were completed over 2 weeks.

Figure 1: the description of the protocol for the sonification group is not clear. As it is currently displayed, it seems that the training comprises two ‘blocks’ of 20 minutes so that each session would total 40 minutes (20 min of gait training + 20 min of acoustic feedback). From what is described in the text, I don’t think that was the case.

Also, it is not clear that each 20-minute session was comprised of 4 blocks including 3 minutes of RTF and 2 minutes of IMS.

I suggest the authors reformulate the figure to clarify the study design.

Line 118: Did patients in the control group receive any information post-intervention about their allocation to a control group, or was there any information in the Consent Form about the study design that would indicate that there were two different test groups?

Participants should be aware of the study design prior to allocation and that they may be assigned to a control condition (unless it is a deception study). Other measures usually taken in randomized studies is to offer participants in the control group the chance to undertake the experimental treatment after the study is completed should they choose.

Please comment on how the allocation information was provided to participants in the control group pre or post-intervention.

Lines 141-151: The authors mentioned that the same sound pattern displayed during the RTF was used to generate the IMS. Please explain how this was done.

Does this mean that participants also heard snow compression sounds during foot-ground contact and xylophone sounds during knee extensions and that these sounds were presented at a fixed tempo?

Please clarify the protocol of the IMS.

Line 151: As previously commented, the authors need to clarify why they chose to display the two different approaches in the same block and not as separate test groups or separate blocks/sessions.

Was the order of the presentation of the two auditory approaches counterbalanced within blocks or sessions?

How the authors accounted for any interference between the two approaches?

Please comment.

Lines 154-155: I suggest the authors to consider labeling TS 1-5 as Week 1 and TS 6-10 as Week 2 to facilitate readability of the text and graphs.

Table 2: I suggest the authors display the table closer to the paragraph where it is cited (example: Line 152).

Line 164: Include a description of how the coefficient of variation is analyzed and how the results are reported (i.e. units).

Line 173: Please clarify to the reader other outcome measures analyzed (example: stride length and stride time).

Results

Lines 186-187: Include statistical results and p-value - even though they are not significant.

Lines 195-212: It seems odd to compare groups and acoustic feedback mode in relation to stride variability if the control group did not receive any acoustic stimuli. The result that there were no significant differences over time in stride length and stride time for the control group in relation to ‘acoustic mode’ is somewhat expected. Therefore, the analysis of the interaction between mode and group seems less informative. Please clarify the reasoning for running this analysis.

Lines 218-219: This sentence is unclear. Please rephrase.

“Though, groups neither differed in temporal nor in mode change regarding these parameters.”

Discussion

Line 223: The authors should not make such a strong statement:

“ … the dual mode gait sonification method improves gait symmetry in patients following unilateral hip arthroplasty”

The authors need to consider that the decrease in step length of the affected leg in the sonification group was not statistically significant and that all the temporo-spatial outcome measures improved from baseline to post-intervention for both groups – even for the control group who did not receive any gait sonification.

Thus, the authors cannot affirm that the changes observed were uniquely linked to the intervention.

I advise the authors to refine the interpretation of the results throughout the discussion and conclusion, taking into consideration all results.

Lines 230-234: Please revisit the interpretation of the findings.

-       Step length of the affected leg showed a trend toward decrease from Week 1 to Week 2 for the sonification group. In the control group, there was a significant increase in step length.

-       There was an increase in step length of the unaffected leg for both groups – not just the sonification group.

-       When comparing the two sonification modes, there was an increase in gait variability for the real-time feedback in Week 2.

Lines 230-233: The hypothesis (H1) stated in the introduction (and here) includes only gait symmetry, not gait variability (as reported in line 233).

Lines 233-234: I can see the analysis comparing the effect of the two sonification modes on gait variability but not the analysis of the interaction between modes and gait symmetry.

Lines 235-238: As mentioned earlier, it is not clear the author’s motivation for combining two different sonification models in the same intervention. Why combining the two models would be different or better than using a single sonification model? 

Lines 244-247: There were also changes in step length in the control group. How do you interpret this result?

Lines 256-258: Would this also apply to the control group?

Lines 270-272: This sentence is confusing. Please rephrase.

Line 273: I don’t think the reference to a study investigating music cues is the most appropriate here since the present study does not use music as cueing for gait. There is a vast literature on rhythmic auditory stimulation that is more relevant for the discussion (see the work of Michael Thaut) – particularly regarding the interpretation that ‘rhythmic auditory cues primarily affect gait speed and correlating parameters, but hardly gait variability’.

Line 283-293: The authors also need to consider that real-time feedback and auditory cues (or instructive feedback) tap into very different underlying mechanisms. Auditory cues presented at a fixed tempo provide crucial anticipatory information that facilitates motor planning and execution (e.g. Thaut et al. 2015). On the other hand, the transformation of dynamic and kinematic movement parameters onto distinct sound components facilitates cross-modal stimulation and sensorimotor representation of the movement to be (re)learned (Effenberg et al., 2016). 

Thaut, M. H., McIntosh, G. C., & Hoemberg, V. (2015). Neurobiological foundations of neurologic music therapy: rhythmic entrainment and the motor system. Frontiers in psychology, 5, 1185.

Effenberg, A. O., Fehse, U., Schmitz, G., Krueger, B., and Mechling, H. (2016). Movement sonification: effects on motor learning beyond rhythmic adjustments. Frontiers in Neuroscience, 10, 219.

Line 307: Authors should also comment on the duration of the intervention (2 weeks) as a potential limitation of the study.

Lines 318-331: Study conclusions need to be revisited considering all study results.